# Effect of Anti-Interleukin-6 Agents on Psychopathology in a Sample of Patients with Post-COVID-19 Syndrome: An Observational Study

**DOI:** 10.3390/brainsci14010047

**Published:** 2024-01-03

**Authors:** Alessio Simonetti, Antonio Restaino, Evelina Bernardi, Ottavia Marianna Ferrara, Stella Margoni, Antonio Maria D’Onofrio, Federica Ranieri, Delfina Janiri, Vincenzo Galluzzo, Matteo Tosato, Georgios D. Kotzalidis, Francesco Landi, Gabriele Sani

**Affiliations:** 1Department of Neuroscience, Section of Psychiatry, Fondazione Policlinico Universitario Agostino Gemelli IRCCS, 00168 Rome, Italy; delfina.janiri@unicatt.it (D.J.); gabriele.sani@unicatt.it (G.S.); 2Menninger Department of Psychiatry and Behavioral Sciences, Baylor College of Medicine, 1977 Butler Blvd., Houston, TX 77030, USA; 3Department of Neuroscience, Section of Psychiatry, Università Cattolica del Sacro Cuore, 00168 Rome, Italy; restainoantonio11@gmail.com (A.R.); evelinabernardi@gmail.com (E.B.); ottaviaferrara@icloud.com (O.M.F.); stella.margoni98@gmail.com (S.M.); antoniomdonofrio@gmail.com (A.M.D.); giorgio.kotzalidis@gmail.com (G.D.K.); 4Department of Psychiatry, University of Campania Luigi Vanvitelli, 80138 Naples, Italy; federicaranieri98@gmail.com; 5Department of Geriatrics, Fondazione Policlinico Universitario A. Gemelli IRCCS, 00168 Rome, Italy; vincenzo.galluzzo@policlinicogemelli.it (V.G.); matteo.tosato@policlinicogemelli.it (M.T.); francesco.landi@unicatt.it (F.L.); 6NESMOS (Neurosciences, Mental Health, and Sensory Organs) Department, Faculty of Medicine and Psychology, Sant’Andrea Hospital, Sapienza Università di Roma, Via di Grottarossa 1035-1039, 00189 Rome, Italy; 7Department of Geriatrics, Università Cattolica del Sacro Cuore, 00168 Rome, Italy

**Keywords:** COVID-19, post-COVID-19 syndrome, Il-6, bevacizumab, sarilumab

## Abstract

Interleukin 6 (IL-6) receptor inhibitors tocilizumab and sarilumab have recently been approved for severe coronavirus disease 2019 (COVID-19). They also affect mood, even though their effect on the post-COVID-19 syndrome-related psychopathology still has to be investigated. The aim of this study was to investigate their effect on psychopathology in a sample of patients with post-COVID-19 syndrome. We included 246 patients (34% female, 66% male) aged 18–75 years who had been hospitalized for COVID. Patients were split into those who received anti-IL-6 receptor agents (Anti-IL-6-R, *N* = 88) and those who did not (Ctrl, *N* = 158). The former group was further split into those receiving tocilizumab (TOC, *N* = 67) and those receiving sarilumab (SAR, *N* = 21). Groups were compared based on clinical characteristics before and during COVID-19 as well as on physical and psychiatric symptoms after COVID-19. Ctrl had less psychiatric and physical symptoms during hospitalization and more post-COVID-19 diarrhea, headache, cough, and dyspnea upon exertion than those receiving IL-6-receptor inhibitors. Ctrl also showed greater difficulties in emotion regulation. These differences were driven by TOC vs. Ctrl, whereas differences between SAR and Ctrl or TOC did not reach significance. IL-6 receptor inhibitors are related to a lower post-COVID-19 illness burden and seem to be effective in emotion regulation. Further research is needed to confirm these findings.

## 1. Introduction

Interleukin 6 (IL-6) is a single-chain glycoprotein encoded by the *IL6* gene [1], which is located on human chromosome 7p21 [2]. This cytokine binds to its cell-surface type I cytokine receptor; the complex consists of the binding CD126 and the signal-transducing gp130 (or CD130) component [3]. As a result, it activates a signal transduction cascade through the Janus kinase (JAK)–Signal Transducer and Activator of Transcription (STAT) pathway [4]. Since its identification, cloning, and sequencing [5], it has received various names according to the cellular system where it was studied before it was realized that the molecule was always the same [6]. IL-6 is mainly produced by macrophages, but it is also produced by dendritic cells, neutrophils, B-cells, selected CD4+ T-lymphocytes, endothelium, fibroblasts, and epithelial cells in response to pathogen-associated molecular patterns binding to pattern recognition receptors, triggering inflammatory cytokine production [7]. It has both proinflammatory and anti-inflammatory actions, depending on concentrations and receptor binding modalities [8]. In particular, trans-signaling is proinflammatory, while classical signaling is anti-inflammatory [9]. IL-6 may cross the blood–brain barrier [10], suggesting its possible involvement in neuroinflammation [11,12] and participation in the pathogenesis of depression [13,14,15] and schizophrenia [16,17]. The acute IL-6 response, like the acute stress response, may help deal with environmental distress, but its persistent production may reflect its inability to wipe out the initial irritant and ensue in disease [6,18]. IL-6 participates in and is central to the “cytokine storm” [19], which is related to the development of severe acute respiratory distress syndrome [9] and is a fatal outcome of coronavirus disease 2019 (COVID-19) [20]. High levels of IL-6, and consequently, C-reactive protein (CRP), are found in post-acute sequelae of COVID-19 (PASQ), also called long COVID or post-COVID-19 syndrome [21]. These high levels reflect the fact that the organism is unable to set off the IL-6–triggered proinflammatory response. Chronic inflammation may explain the physical and neuropsychiatric symptoms of long-COVID, whose manifestations and duration are very heterogeneous. Common symptoms include fatigue, headache, respiratory symptoms, i.e., breathlessness, persistent cough, and chest pain, fever, diarrhea, joint pain, myalgia, anosmia, dysosmia, vertigo, and neuropsychiatric symptoms like insomnia, anxiety, depression, and symptoms of post-traumatic stress disorder (PTSD) as well as memory and concentration problems. Their duration varies from weeks to months after recovering from SARS-CoV-2 infection [22].

The implication of IL-6 in PASC prompted investigators to develop and propose the use of specific anti-IL-6 antibodies [14,23]. Tocilizumab, a humanized monoclonal IgG1 antibody that binds and blocks IL-6 receptors (both classical and trans signaling), was developed in the last years of the 20th century in Japan using recombinant DNA technology to treat autoimmune diseases like rheumatoid arthritis [24]. It was introduced in Japan in 2005 for Castleman’s disease, in 2008 to treat rheumatoid arthritis and juvenile idiopathic arthritis, in 2009 in the European Union, and in 2010 in the United States of America [25]. In Europe, it is approved for use in rheumatoid arthritis, systemic juvenile idiopathic arthritis, juvenile idiopathic polyarthritis, giant cell arteritis, and cytokine release syndrome. On 6 December 2021, it received approval recommendations for patients “with severe COVID-19 who required extra oxygen or mechanical ventilation and had high levels of C-reactive protein in the blood” [26]. On 21 December, 2022, the Food and Drug Administration (FDA) “approved Actemra” (tocilizumab) “for the treatment of COVID-19 in hospitalized adult patients who are receiving systemic corticosteroids and require supplemental oxygen, non-invasive or invasive mechanical ventilation, or extracorporeal membrane oxygenation (ECMO)”. “Actemra is also FDA-approved to treat: Adult patients with moderately to severely active rheumatoid arthritis who have had an inadequate response to one or more disease-modifying anti-rheumatic drugs (DMARDs); Adult patients with giant cell arteritis; Slowing the rate of decline in pulmonary function in adult patients with systemic sclerosis associated interstitial lung disease (SSc-ILD); Patients 2 years of age and older with active polyarticular juvenile idiopathic arthritis (PJIA); Patients 2 years of age and older with active systemic juvenile idiopathic arthritis (SJIA); Adults and pediatric patients 2 years of age and older with chimeric antigen receptor (CAR) T cell-induced severe or life-threatening cytokine release syndrome (CRS)”. The FDA has determined Actemra is safe and effective for these uses when used in accordance with the FDA approved labeling” [27].

Sarilumab, another human monoclonal IgG1 antibody that blocks both soluble and membrane-bound IL-6 receptors, hence blocking both classical cis- and trans-signaling, has been developed conjointly by an American and a French pharmaceutical company during the first decade of the century to treat rheumatoid arthritis. Trials started in 2007, [28] and applications for approval were received in 2016 in Japan and 2017 in the US and Europe. On 21 April 2017, the European Medicines Agency (EMA) recommended that the use of sarilumab “in combination with methotrexate (MTX) is indicated for the treatment of moderately to severely active rheumatoid arthritis (RA) in adult patients who have responded inadequately to, or who are intolerant to one or more disease modifying anti rheumatic drugs (DMARDs)” …It “can be given as monotherapy in case of intolerance to MTX or when treatment with MTX is inappropriate” [29]. The FDA approved sarilumab on 22 May 2017 after a dispute and successful trials [30,31,32]. Sarilumab is used as an add-on treatment in adult patients with rheumatoid arthritis who do not respond adequately to disease-modifying agents, MTX or tumor necrosis factor-α inhibitors [33]. It has also been used in polymyalgia rheumatica [34], noninfectious uveitis and juvenile idiopathic arthritis, even though it did not receive approval for these disorders [28]. Differently from tocilizumab, sarilumab did not show consistent effectiveness in treating patients with severe Coronavirus Disease of 2019 (COVID-19). Even though a systematic review made by Chamlagain et al. demonstrated that sarilumab represents a safe and potentially useful tool in patients with COVID-19 [35]; three recent studies showed no differences with comparators or placebo [36,37,38]. The EMA and the FDA hesitate to include sarilumab among drugs recommended for patients with COVID-19, but the World Health Organization (WHO) issued a “strong recommendation to use IL-6 receptor blockers (tocilizumab or sarilumab) in patients with severe or critical COVID-19” on 6 July 2021 [39]. The Italian Drugs Agency [40] recommended sarilumab in patients with severe COVID-19 when tocilizumab is unavailable.

While IL-6-receptor antagonists are expected to improve depression and mood in general in patients with a mood disorder, given that they counteract the neuroinflammation that IL-6 is likely to induce [41], they are still to be tested in patients with major depressive disorder. For the moment, there is a protocol for an Oxford (UK) study proposed since 2018 [42] that has still to provide results and another study in ClinicalTials.gov (NCT02660528) that is being conducted in Boston, Massachusetts, US at the Brigham and Women’s Hospital by Jessica Harder. This study has partial results. Other studies focusing on mood during the administration of tocilizumab in patients with rheumatoid arthritis found reductions in depression ranging from small effect sizes [43] to significant, correlating with [44] or disjoint from improvement in arthritis symptoms [45]. However, one study conducted in patients undergoing allogeneic hematopoietic stem cell transplantation found tocilizumab to be significantly more associated with mood worsening after four weeks compared to absence of treatment [46]. Another study was carried out in patients with moderate to severe COVID-19-related pneumonia after their discharge. Subjects receiving tocilizumab in adjunct to the standard treatment showed marginally worse levels of depression than those undergoing the standard treatment only at the 3-month follow-up, whereas an absence of depression was observed at the 6-month follow-up [47]. Sarilumab has not been assessed for its effects on depression, but one study on subjects with rheumatoid arthritis showed more effectiveness on depression as compared to adalimumab [48]. Taken together, results suggest that IL-6 receptor inhibitors are likely to have effects on mood, but the direction of these effects have not been defined.

In this study, we selected from our cohort of patients who had been hospitalized for COVID-19, those patients meeting requirements for anti-Il-6 treatment. We divided the sample according to the anti-Il-6 treatment assumed in addition to the as-usual schedule, consisting of corticosteroids and heparin. Therefore, groups were composed by subjects receiving tocilizumab or sarilumab, when tocilizumab was unavailable, or treated them according to an as-usual schedule. The latter composed the control group (Ctrl). Subjects receiving tocilizumab and sarilumab were grouped according to the drug they received and also grouped together to form the IL-6 receptor inhibitor group. The groups were compared for somatic PASC and psychological effects cross-sectionally. Our aim was to see whether administering IL-6 receptor inhibitors in patients who had persistent COVID-19 symptoms and sequelae was related to psychological or somatic symptoms.

## 2. Materials and Methods

The study was conducted by the Gemelli Against COVID-19 Post-Acute Care Study Group of the Fondazione Policlinico Universitario “Agostino Gemelli” IRCCS of Rome, Italy. This study’s sample is part of a larger sample recruited in the Post-Acute Care Service at Fondazione Policlinico Universitario “Agostino Gemelli” IRCCS, Rome, Italy. Subjects enrolled were those who had COVID-19 and tested negative. In the aforementioned service, patients underwent a multidisciplinary assessment, including internal medicine, geriatric, ophthalmological, otolaryngologic, pneumological, cardiological, immunological, rheumatological and psychiatric evaluations. The enrollment started on 21 April 2020 and ended on 11 May 2022. As regards this study, the following criteria were applied: (a) age between 18 and 75 years; (b) previously testing positive for COVID-19; (c) hospitalization for COVID-19; (d) capability of providing informed consent; (e) presence of a post-COVID-19 syndrome. Exclusion criteria were: (a) severe neurodevelopmental disorders, (b) dementia or other severe neurological disorders. Considering the lack of homogeneity in the proposed criteria for the definition of the post-COVID-19 syndrome [49], we defined it as any condition characterized by the presence of one or more symptoms that appeared or persisted after testing negative, regardless of symptom duration or severity, in the absence of any external explanation unrelated to COVID.

Demographic characteristics, medical history, severity and course of COVID-19 infection were first collected by an internal medicine specialist. As regards psychiatric evaluation, one or more psychiatrists from the aforementioned study group performed an interview in which psychiatric history was collected. During this interview, detailed information regarding the presence and characteristics of psychiatric history were collected possibly with the aid of the subjects’ treating psychiatrist. Also, interviews with subjects’ relatives were also performed in order to clarify patients’ psychopathology. Rating scales assessing severity of general psychopathology, severity of depressive symptoms, severity of anxiety symptoms, and severity of post-traumatic symptoms were also administered. Furthermore, patients were asked to complete self-questionnaires assessing severity of mixed depression, resilience, insomnia, emotional dysregulation, pleasure, and hopelessness.

Recruited patients gave their written informed consent to participate in the study prior to study initiation, which was approved by the Ethical Committee of the Fondazione Policlinico Universitario Agostino Gemelli IRCCS (protocol ID number: 0013008/20, date of approval: 30 August 2020).

### 2.1. Assessment

Patient data considered for the present study were: (a) sociodemographic and clinical characteristics (i.e., age, gender, marital status, occupation, education, and body mass index (BMI)); (b) medical history prior to COVID-19, i.e., number of clinical comorbidities (including neurological, cardiological, and systemic comorbidities), presence/absence pneumological comorbidities, and number of drugs prescribed and taken; (c) data regarding SARS-CoV-2 infection, i.e., type of treatments received prior to hospitalization, type and number of symptoms during the infection, length of hospitalization, use of oxygen therapy, presence/absence of intensive care unit (ICU) hospitalization; type and number of drugs assumed during hospitalization; (d) data regarding a post-COVID-19 syndrome, i.e., number and type of symptoms experienced after testing negative for COVID-19, peripheral capillary oxygen saturation (SpO_2_), time elapsed between COVID-19 and evaluation in the Post-Acute Care Service; (e) data regarding psychiatric evaluation, i.e., number of psychotropic drugs assumed prior to COVID-19, psychiatric symptoms and rates of assumption of psychotropic drugs during hospitalization, and the presence of psychiatric symptoms during the post-COVID-19 syndrome as assessed with rating scales. Specifically, the following domains were investigated through the scales: severity of general psychopathology with the Brief Psychiatric Rating Scale (BPRS), severity of depressive symptoms with the Hamilton Rating Scale for Depression (HAM-D), severity of anxiety symptoms with the Hamilton Anxiety Rating Scale (HAM-A), severity of mixed depression with the Koukopoulos Mixed Depression Rating Scale (KMDRS), severity of post-traumatic stress disorder symptoms with the Clinician-Administered PTSD Scale, resilience with the Connor–Davidson Resilience Scale (CD-RISC), insomnia and other sleep problems with the Pittsburg Sleep Quality Index (PSQI), emotion dysregulation with the Difficulties in Emotion Regulation Scale (DERS), pleasure with the Snaith–Hamilton Pleasure Scale (SHAPS), and hopelessness with the Beck Hopelessness Scale (BHS). All used psychiatric rating scales are described below.

The 24-item Brief Psychiatric Rating Scale (BPRS) [50] is a widely used clinical assessment tool designed to assess the severity of general psychopathology. This scale evaluates the severity of a wide variety of psychiatric symptoms, including anxiety, depression and psychosis. Psychiatric symptoms are rated from 1 (not present) to 7 (extremely severe). The total score varies from 24 to 168, where lower scores indicate less severe psychopathology. We used here the approved and validated Italian version [51].

The Hamilton Rating Scale for Depression (HAM-D) [52] assesses severity of depressive symptoms. The 17-variables version was used in the present study. Each variable explores depressive symptoms experienced over the past week. Each item is scored on a scale from 0 (absent) to 2 or 4 (severe), depending on the specific item. The total score is calculated by summing the scores for all the items. A total score of 0–7 indicates the absence of depression; scores of 8–16 suggest mild depression; scores of 17–23 indicate moderate depression and scores over 24 are indicative of severe depression.

The Hamilton Anxiety Rating Scale (HAM-A) [53] assesses anxiety. It consists of 14 items. Items 7–13 explore somatic anxiety, while psychic anxiety is explored with items 1–6 and 14. Each item is scored on a scale from 0 (absent) to 4 (severe). The total score, calculated by summing each item score, varies from 0 to 56. Scores <17 indicate mild anxiety, 18–24 indicate mild to moderate, 25–30 indicate moderate to severe, and >30 indicate severe anxiety.

The Clinician-Administered PTSD Scale (CAPS) [54] is a structured interview designed to evaluate both the diagnostic status and the severity of symptoms related to post-traumatic stress disorder (PTSD). Along with examining the symptoms outlined in the DSM-5 for PTSD, the questions focus on the onset and duration of symptoms, subjective distress, the impact of symptoms on social and occupational functioning, improvement observed since the last CAPS assessment, overall response validity, overall PTSD severity, and additional features like dissociative symptoms (depersonalization and derealization). DSM-5 criteria-based PTSD symptoms are subdivided into 5 clusters. The CAPS-5 total symptom severity score is computed by summing the severity scores for all DSM-5 PTSD symptoms with each symptom’s severity scored on a scale from 0 (absent) to 4 (extremely/incapacitating). Likewise, the severity scores for individual items within the same DSM-5 cluster are summed to determine the CAPS-5 symptom cluster severity scores. Additionally, a symptom cluster score for dissociation can be obtained by summing items 19 and 20.

The Koukopoulos Mixed Depression Rating Scale (KMDRS) [55] is a self-administered rating scale. The KMRS consists of 14 items assessing the presence and severity of a variety of symptoms related to mixed depression, involving mood, energy level, irritability, and others. Possible scores range from 0 to 51. Higher scores indicate greater severity of mixed depressive symptoms.

The Connor–Davidson Resilience Scale (CD-RISC) [56] is a self-administered rating scale assessing resilience. It consists of 25 items which are evaluated on a five-point Likert scale ranging from 0 to 4. The total scores vary between 0 and 100; higher scores reflect higher resilience, and an increase is considered an improvement.

The Pittsburg Sleep Quality Index (PSQI) [57] is a self-assessment questionnaire composed of 7 components including subjective sleep quality, sleep latency, sleep duration, sleep efficiency, sleep disturbance, use of sleep medication, and daytime dysfunction. In scoring the PSQI, the components score from 0 (no difficulty) to 3 (severe difficulty). Then, the component scores are summed to produce a global score (range 0 to 21). Higher scores indicate worse sleep quality.

The Beck Hopelessness Scale (BHS) [58] is a self-assessment questionnaire. It consists of 20 items covering three domains: feelings about the future, loss of motivation and future expectations. Respondents indicate whether each statement applies to them by selecting “true” or “false”. The total BHS score varies from 0 to 20, with higher scores reflecting higher levels of hopelessness. Specifically, total scores of 0–3 are not pathological, while 4–8 suggest mild hopelessness, 9–14 suggest moderate hopelessness and scores greater than 14 indicate severe hopelessness.

The Difficulties in Emotion Regulation Scale (DERS) [59] is a self-assessment questionnaire measuring emotion regulation problems. The 36 items explore several components: nonacceptance of emotional responses, difficulty engaging in goal-directed behavior, impulse control difficulties, lack of emotional awareness, limited access to emotion regulation strategies, lack of emotional clarity. This scale explores not just the modulation of emotional arousal but also the awareness, understanding, and acceptance of emotions, and the ability to act in desired ways regardless of emotional state. We used the validated Italian version [60]. Higher scores indicate more difficulty in emotional regulation.

The Snaith–Hamilton Pleasure Scale (SHAPS) [61] is a self-administered rating scale assessing anhedonia. It consists of 14 items in which patients are asked to indicate their level of agreement or disagreement with statements related to experiencing pleasure over the last few days. Each item is scored as “definitely agree”, “agree”, “disagree”, and “strongly disagree”. The first two receive 0 points, the last two receive 1. There are reverse items, i.e., items 2, 4, 5, 7, and 9. SHAPS scores range from 0 to 14. If someone agreed or strongly agreed with every statement, their score would be 0. Higher SHAPS scores indicate higher levels of anhedonia. We used the validated Italian version of the instrument [62].

Subjects enrolled for the present study were further divided according to the presence or absence of IL6-R drug treatment assumption. Some of them actually received anti-IL6-R drugs (tocilizumab or sarilumab) and some of them did not; the latter constituted the control group (Ctrl). The Italian Drug Agency (AIFA) indications for the use of tocilizumab and sarilumab in COVID-19 are: (a) patients admitted to the intensive care unit for less than 24/48 h who received mechanical ventilation or high-flow oxygen; patients recently admitted with rapidly increasing oxygen requirement, needing noninvasive mechanical ventilation or high-flow oxygen, and having high levels of inflammation (C-reactive protein (CRP) ≥ 75 mg/L); and (b) patients experiencing rapid clinical deterioration after 24/48 h of dexamethasone or other corticosteroid use [40,63]. However, tocilizumab has been approved for treatment of COVID-19 on 9 June 2021, while treatment guidelines for sarilumab received approval on 22 September 2022. Prior to anti-IL-6-R approval, tocilizumab and sarilumab were used off-label. Although their indications were unclear, many studies used anti-IL-6-R in patients with severe or critical COVID-19 [64,65,66,67]. Since the enrollment of our study started on April 2020, in agreement with the aforementioned studies, we considered enrollable for our study all patients with severe or critical COVID-19 who received anti-IL-6-R during hospitalization. Therefore, the sample was divided into patients who received treatment with anti-IL-6-R during hospitalization for COVID-19 (Anti-IL-6-R), and those who did not (Ctrl). The first group was further divided according to the specific anti-IL-6-R drug received, i.e., intravenous tocilizumab during hospitalization (TOC) and intravenous sarilumab during hospitalization (SAR). Information regarding administration of sarilumab and tocilizumab was further collected through a detailed review of patients’ medical records and, when available, with the contribution of each patient’s responsible physician during hospitalization.

### 2.2. Statistical Analysis

A Kolmogorov–Smirnov test was preliminary performed for all continues variables in order to verify the suitability of *t*-tests/ANOVAs. Since all the variables were not normally distributed, differences between Anti-IL-6-R and Ctrl regarding continuous variables were analyzed with the Mann–Whitney U test. The chi-squared test (χ2) was used for nominal variables.

In each Mann–Whitney U/χ2-tests, the groups (Ctrl; Anti-IL-6-R; TOC; SAR) were independent variables, while sociodemographic characteristics (age, gender, occupation, educational level), medical history data (clinical, and in particular respiratory comorbidities, calculated as number of total comorbidities and presence/absence of respiratory comorbidities, and type and number of drugs assumed at intake), data regarding SARS-CoV-2 infection and the post-COVID-19 syndrome (time elapsed from COVID-19 onset, type and number of symptoms present during COVID-19, hospitalization length, number and type of symptoms during the post-COVID-19 syndrome, and number and types of current medications), data regarding psychiatric evaluation (presence/absence of psychiatric history, number of psychotropic drugs during hospitalization, scores on psychopathological scales), were dependent variables. Correlations among psychopathological scales were explored with Pearson correlations.

Differences between TOC, SAR and Ctrl were performed through χ2-tests for discrete variables and the Kruskall–Wallis test for continuous variables. In each Kruskall–Wallis/χ2-test, the three groups (Ctrl; TOC, SAR) were independent variables, while sociodemographic characteristics (age, gender, occupational status, educational level) and data regarding medical history (as above) were dependent variables. Mann–Whitney U tests were used as post hoc tests for continuous variables. As regards the results of the χ2-tests, post hoc testing was performed by Z-tests for independent proportions.

We also investigated the possible effects of confounding variables. Results of Kruskall–Wallis and *t*-tests were corrected for the effect of demographic and clinical variables showing significant differences among groups through multiple ranked one-way analyses of covariance (ANCOVAs), also known as Quade’s ANCOVA. In each ranked ANCOVA, variables showing differences among groups entered the model. We used the statistical routines of SPSS Statistics 24.0 for Windows (IBMCo., Armonk, NY, USA, 2016).

## 3. Results

A total of 1586 patients were enrolled. After applying criteria for the present study, the final sample consisted of 246 patients. All the subjects have been hospitalized prior to the availability of vaccines. Patients had a mean age of 56.74 years and were mostly men (*n* = 162, 65.85%); most of them (*n* = 179, 72.8%) had an occupation and they had a mean of 13.14 years of education. Table 1 shows the sociodemographic and baseline clinical characteristics of the Ctrl and Anti-IL-6-R groups. Ctrl showed a higher rate of respiratory comorbidity. As regards differences in data related to SARS-CoV-2 infection and the post-COVID-19 syndrome, Ctrl showed a higher frequency of headache, asthenia, arthralgia, syncope, and Raynaud’s phenomenon. Furthermore, Ctrl had more patients who required oxygen therapy during acute COVID-19 infection. Also, they had more post-COVID-19 syndrome’s symptoms at the time of evaluation: namely, diarrhea, headache, and exertional dyspnea (Table 1 and Table 2).

Differences regarding psychiatric symptoms are shown in Table 2, whereas exploratory correlations were described in Appendix A. Ctrl scored higher on the DERS. Regarding all other scales, no significant differences emerged.

Among sociodemographic differences, only sex differed between TOC, SAR and Ctrl, as shown with Kruskall–Wallis tests. The post hoc test showed that there were less women in the SAR group compared with Ctrl and TOC (Table 3).

Regarding COVID-19-related variables, significant differences were related to cortisone therapy prior to hospitalization, oxygen therapy during hospitalization and presence of COVID-19-related fatigue, headache, cutaneous and mucosal lesions. Post hoc tests showed higher rates of patients who underwent cortisone therapy and oxygen therapy in Ctrl compared to SAR. Fatigue was more frequent in Ctrl than TOC; headache was more frequent in Ctrl than SAR; cutaneous and mucosal lesions were more frequent in Ctrl than both SAR and TOC (Table 3).

Concerning post-COVID-19 syndrome’s symptoms, significant differences were shown in rates of dyspnea on exertion, whereas rates of headache and cough approached significance. Post hoc analyses showed that dyspnea and headache were more frequent in Ctrl than in SAR, whereas cough was more frequent in Ctrl than in TOC (Table 4). Regarding post-COVID psychopathological evaluation, we found a significant difference in DERS total scores with Ctrl scoring higher than TOC in post hoc analyses (Table 4 and Figure 1).

### Effects of Possible Confounding Variables

The effect of possible confounding variables was limited to DERS scores. Concerning the comparison between Ctrl and Anti-IL-6-R, respiratory comorbidities, headache, asthenia, arthralgia, syncope, Raynaud’s phenomenon, oxygen therapy, post-COVID diarrhea, post-COVID headache, post-COVID cough and post-COVID dyspnea on exertion entered the ranked ANCOVA. As for the three-group comparisons (Ctrl vs. TOC vs. SAR), results were adjusted for sex, cortisone therapy, oxygen therapy, fatigue, headache, cutaneous/mucosal lesions, and post-COVID dyspnea on exertion. In both cases, results remained significant.

## 4. Discussion

In this study, we found that compared to controls, patients receiving IL-6 receptor inhibitors had more psychiatric symptoms or delirium during hospitalization and less pulmonary conditions, weakness, joint pain, dysautonomia, fainting, and need for oxygen therapy at baseline. They also displayed less post-COVID-19 diarrhea, headache, cough and dyspnea upon exertion (this was largely due to control vs. tocilizumab differences); the only symptom of psychiatric relevance that differentiated patients receiving IL-6 receptor inhibitors from controls was emotion dysregulation, which was higher in the control group (this was mainly accounted for by control vs. tocilizumab differences). We found no differences between the two groups in depression, anxiety, general psychopathology, ability to perceive pleasure, hopelessness, mixed mood symptoms, or post-traumatic symptoms.

When it comes to comparisons between controls and each of the IL-6 receptor inhibitors, there were less women in the sarilumab group than in the control or the tocilizumab groups. This might have to do with the dearth of patients who entered the sarilumab group (just 21). In turn, the restricted number of patients who received sarilumab was due to regulatory recommendations, which rendered tocilizumab unavailability a precondition for sarilumab administration. Controls had more mucocutaneous lesions than both tocilizumab and sarilumab patients, while they had more muscle aches and weakness than patients in the tocilizumab group and more cortisone prior to admission, need for oxygen therapy, and headaches than patients in the sarilumab group. The two IL-6 receptor inhibitors did not differ from one another on any measure.

Given the smallness of the sarilumab sample, we may not speculate as to the differences with the control group, as most of IL-6 receptor inhibitor-control differences could be attributed to tocilizumab vs. controls differences. We investigated side effects and somatic symptoms of the PASC/post-COVID-19 syndrome and found differences in side effects and emotional regulation difficulties between patients who did and who did not receive the IL-6 receptor inhibitor but not between patients who received one or the other IL-6 receptor inhibitor. Tocilizumab and sarilumab have been tested for safety and found to be similar in a study comparing intravenous tocilizumab with subcutaneous sarilumab [68]. We here administered both IL-6 receptor inhibitors intravenously and obtained similar results regarding safety. Switching from intravenous to subcutaneous dosing produces no additional safety concerns [69]. In patients with COVID-19, both tocilizumab and sarilumab improved outcome measures, but tocilizumab showed more promise in reducing mortality at four weeks and progression to the need for mechanical ventilation in patients with moderate-to-severe COVID-19 [70], which is something that a further network meta-analysis [71] and a systematic review and metanalysis [72] could not demonstrate. The two drugs were found to be of similar potency in reducing 60-day mortality in patients with severe COVID-19 with the lower tocilizumab dose (8 mg/kg) showing the best results [73], while in another Bayesian network meta-analysis, both drugs were effective at reducing the mortality of severe and critical COVID-19 patients when used in combination with corticosteroids [74]. We were not able to conduct a balanced comparison between tocilizumab and sarilumab due to the fact that the administration of the latter is subordinated to the unavailability of the former. A shortage of tocilizumab was one reason why patients in the Swets et al. [73] study were switched to sarilumab.

Tocilizumab costs about half as much as sarilumab [75]. The potency of suppression of IL-6/STAT3 signaling differed between tocilizumab and sarilumab, with the latter being stronger than the former in vivo at 200 mg biweekly subcutaneous injections vs. 162 mg biweekly subcutaneous injections, but it was weaker than 162 mg weekly tocilizumab subcutaneous injections [76]. In vitro, stronger receptor occupancy and C-reactive protein reduction were found for sarilumab than for tocilizumab [77]. Combining all these data, we have the perception that tocilizumab is rightly promoted as the one IL-6 receptor inhibitor to use in severe COVID-19 and the post-COVID syndrome with sarilumab constituting a valid alternative in case of unavailability of tocilizumab.

We found more emotional dysregulation in the sample that received treatment-as-usual without taking any IL-6 receptor inhibitor. Emotional dysregulation has been linked to the pandemic [78,79], but it appears to mediate the relationship between loneliness and depression [80,81], while in another study, emotional dysregulation and reduced hedonic tone predicted depressive symptoms [82]. Another study during the pandemic found emotional dysregulation to mediate the relationship between trauma and substance use disorders [83]. Mechanisms linking emotional dysregulation, SARS-CoV2 infection and IL-6 are still not known. Nevertheless, IL-6 and its receptors are abundantly expressed in all the anatomic areas involved in emotion regulation, such as the amygdala, the hippocampus, the hypothalamus, the habenula and the piriform cortex [84,85], and exposure to various stressors has been related to increased plasma levels of IL-6 [86]. The injection of IL-6 in the amygdala induces depression-like behaviors [87], and mice hyper-expressing IL-6 showed increased excitability of the central nucleus of the amygdala [88]. The central nucleus of the amygdala is particularly important for mood regulation. It integrates cortical, brainstem, and intra-amygdala afferents to coordinate behavioral and physiologic responses via GABAergic projections to downstream “effector” regions. These regions included the periaqueductal gray, the lateral and paraventricular hypothalamus, the locus coeruleus and the dorsal vagal complex, as well as the prefrontal cortex [89]. Activation of these areas is responsible for the behavioral effect of emotion [89]. The dysregulation of GABAergic central amygdala interneurons has been reported in mice overexpressing IL-6 [88]. Therefore, the hyperexpression of IL-6-related emotional dysregulation might act via central amygdala GABAergic dysfunctions. Thus, IL-6 inhibition might balance mood through amygdala GABAergic modulation. However, we found better emotional regulation in subjects under tocilizumab and not in those with sarilumab. To this extent, in vitro studies showed that sarilumab binds the IL-6 receptor with 15- to 22-fold higher affinity than tocilizumab [77,90]. Mood dysregulation has been associated either with increased or decreased levels of IL-6 [91,92]. Therefore, we might speculate that sarilumab might excessively inhibit central amygdala IL-6 levels, thus resulting in reduced emotional regulation ability [93].

### Limitations

This study has several limitations. The main limitation is its cross-sectional design, which did not allow us to assess the included patients’ psychological status prior to inclusion in the study. For the same reason, it was not possible to clearly define the timeframe between the psychopathological assessment and the exact day of COVID-19 negativity as confirmed by negativity of a COVID-19 test. Furthermore, sample sizes were relatively small, and the sarilumab sample was much smaller than the tocilizumab sample. This was due to the fact that the AIFA recommends that sarilumab should be given in severe COVID-19 patients in need of respiratory aid only when tocilizumab is not available. Tocilizumab shortage is not all that frequent; this resulted in a lack of balance of the two drug groups. Furthermore, it turned out that more subjects in the Ctrl group required oxygen therapy during acute COVID-19 infection. This could have affected the psychological characteristics of the group receiving IL-6 receptor inhibitors.

## 5. Conclusions

Our results are compatible with the use of adjunct IL-6 receptor inhibitors in cases of severe to critical COVID-19 and PASC; both tocilizumab and sarilumab appear to be fit.

Subjects that received adjunctive IL-6 receptor inhibitors showed lower COVID-19-related symptoms than those receiving the standard, as-usual treatment. As regards psychiatric measures, a better emotional regulation was observed in subjects receiving tocilizumab. This might suggest a possible future application of tocilizumab in the treatment of disorders in which dysregulated mood represents a cardinal feature, such as major depressive disorder, bipolar disorder, or borderline personality disorder [94]. Due to the insufficient number of data regarding the use of tocilizumab in these disorders, future research is warranted.

## Figures and Tables

**Figure 1 brainsci-14-00047-f001:**
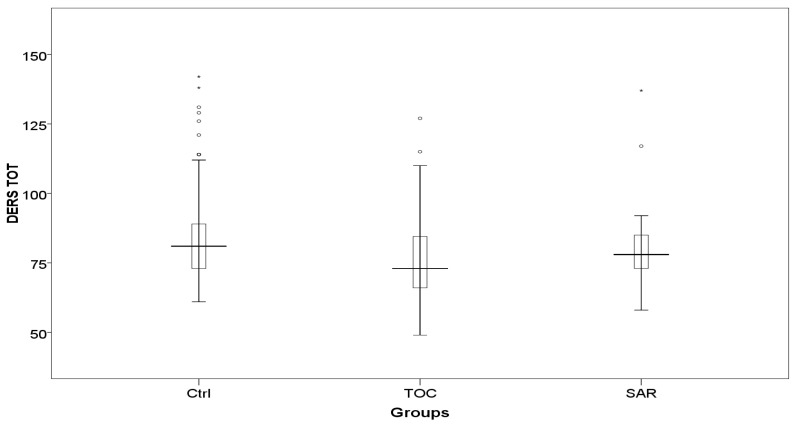
DERS total scores in TOC, SAR, and Ctrl. Legend: Ctrl, patients who did not receive anti-interleukin-6-receptor treatment; DERS: Difficulties in Emotion Regulation scale; SAR, patients receiving Sarilumab during hospitalization; TOC, patients receiving tTocilizumab during hospitalization. * *p* < 0.05, ** *p* < 0.01.

**Table 1 brainsci-14-00047-t001:** Differences in sociodemographic, clinical history prior to COVID-19, and COVID-19 parameters between Anti-IL-6-R and Ctrl.

	Ctrl (*N* = 158)	Anti-IL-6-R (*N* = 88)	*U* or χ^2^	*p*-Value
**Sociodemographic and clinical characteristics**				
Age (y), mean ± SD	56.66 ± 10.30	56.90 ± 12.12	6640	0.56
Female, *n* (%)	58 (36.7)	26 (29.5)	1.29	0.26
Marital Status (%)				
Never married	18 (11.5)	15 (17)	4.94	0.29
Married	101 (63.9)	61 (69.3)		
Partner	7 (4.4)	2 (2.3)		
Divorced	7 (4.4)	2 (2.3)		
Widowed	25 (15.8)	8 (9.1)		
Occupation (%)				
Unemployed	9 (5.6)	7 (7.9)	1.85	0.60
Employed	117 (74.1)	62 (70.5)		
Retired	32 (20.3)	19 (21.6)		
BMI, mean ± SD	28.0 ± 4.57	28.23 ± 4.65	6597	0.66
Education (y), mean ± SD	12.80 ± 3.93	13.76 ± 4.13	5723	0.07
**Clinical history prior to COVID-19**				
Comorbidities prior to COVID-19, mean ± SD	2.69 ± 2.40	2.36 ± 2.30	6292	0.21
Psychiatric treatments prior to COVID-19, mean ± SD	0.07 ± 0.30	0.10 ± 0.37	6793	0.49
**Pneumological comorbidities, *n* (%)**	31 (19.6)	8 (9.1)	4.70	**0.03**
Antibiotics prior to admission	89 (56.3)	52 (59.1)	0.18	0.67
Cortisone prior to admission	67 (42.4)	26 (29.5)	3.97	0.05
Drug treatments prior to admission (n), mean ± SD	1.27 ± 1.64	1.13 ± 1.67	6206	0.15
**COVID-19**				
**Psychiatric symptoms/delirium during hospitalization, *n* (%)**	8 (5.1)	11 (12.5)	6435	**0.04**
Psychiatric treatments during hospitalization (n), mean ± SD	0.17 ± 2.33	0.18 ± 1.97	6880	0.82
Number of total symptoms during COVID-19, mean ± SD	7.59 ± 3.96	6.76 ± 3.44	6086	0.11
Fever, *n* (%)	142 (89.9)	81 (92.0)	0.31	0.57
Cough, *n* (%)	97 (61.4)	56 (63.6)	0.12	0.73
**Asthenia/fatigue/weakness, *n* (%)**	133 (84.7)	62 (70.5)	7.06	**<0.01**
Diarrhea, *n* (%)	40 (25.3)	16 (18.2)	1.64	0.20
**Headache, *n* (%)**	81 (51.3)	30 (34.1)	6.73	**0.01**
Anosmia/Dysosmia, *n* (%)	62 (39.2)	39 (44.3)	0.60	0.44
Dysgeusia, *n* (%)	65 (41.1)	40 (45.5)	0.43	0.51
Red eyes, *n* (%)	27 (17.1)	14 (15.9)	0.06	0.81
Reduction in eyesight, *n* (%)	25 (15.8)	12 (13.6)	0.21	0.65
Reduction in eyesight, *n* (%)	15 (9.5)	7 (8.0)	0.16	0.68
**Syncope, *n* (%)**	46 (29.1)	15 (17.0)	4.41	**0.04**
Vertigo, *n* (%)	95 (60.1)	42 (47.7)	3.52	0.06
**Arthralgia, *n* (%)**	25 (15.8)	4 (4.5)	6.91	**0.01**
Cutaneous and mucosal lesions, *n* (%)	38 (24.1)	14 (15.9)	2.25	0.13
Sicca Syndrome, *n* (%)	2 (1.3)	0 (0.0)	1.12	0.29
**Raynaud’s phenomenon, *n* (%)**	95 (60.1)	39 (44.3)	5.69	**0.02**
Myalgia, *n* (%)	56 (35.4)	25 (28.4)	1.27	0.26
Sore throat, *n* (%)	35 (22.2)	21 (23.9)	0.09	0.76
Sputum, *n* (%)	32 (20.3)	17 (19.3)	0.03	0.86
Rhinitis, *n* (%)	20 (12.7)	14 (15.9)	0.50	0.48
Loss of appetite, n (%)	69 (43.9)	47 (53.4)	2.02	0.15
Days of hospitalization, mean ± SD	22.35 ± 25.10	26.47 ± 19.83	1.76	0.19
**Oxygen therapy, *n* (%)**	141 (89.2)	69 (78.4)	5.31	**0.02**
Drugs during hospitalization (*n*), mean ± SD	2.44 ± 2.54	2.06 ± 2.39	1.32	0.25
Intensive Care Unit admission, *n* (%)	113 (71.5)	57 (60.8)	1.20	0.27

Legend: significant results are in bold. Abbreviations: Anti-IL-6-R: patients who received anti-interleukin-6-receptor treatment; Ctrl: patients who did not receive anti-interleukin-6-receptor; SD, standard deviation.

**Table 2 brainsci-14-00047-t002:** Differences in clinical variables and scores on psychopathological scales at post-COVID-19 evaluation between Ctrl (*N* = 158) and Anti-IL-6-R (*N* = 88).

	Ctrl	Anti-IL-6-R	*U* or χ^2^	*p*
**Number of total post-COVID-19 symptoms, mean ± SD**	3.88 ± 2.78	3.23 ± 2.63	5714	**0.02**
Asthenia/fatigue Post-COVID, *n* (%)	102 (64.6)	50 (56.8)	1.43	0.23
**Cough Post-COVID, *n* (%)**	28 (17.7)	6 (6.8)	5.64	**0.02**
**Diarrhea Post-COVID, *n* (%)**	7 (4.4)	0 (0)	4.01	**0.04**
**Headache Post-COVID, *n* (%)**	35 (22.2)	9 (10.2)	5.47	**0.02**
Anosmia/dysosmia Post-COVID, *n* (%)	17 (10.8)	9 (10.2)	0.02	0.90
Dysgeusia Post-COVID, *n* (%)	22 (13.9)	7 (8.0)	1.94	0.16
Red eyes Post-COVID, *n* (%)	8 (5.1)	5 (5.7)	0.04	0.83
Reduction in eyesight Post-COVID, *n* (%)	26 (16.5)	16 (18.2)	0.12	0.73
Syncope Post-COVID, *n* (%)	0 (0)	0 (0)	–	–
Vertigo Post-COVID, *n* (%)	22 (13.9)	7 (8.0)	1.94	0.16
Arthralgia and arthritis Post-COVID, *n* (%)	62 (39.2)	31 (35.2)	0.39	0.53
Cutaneous and mucosal lesions Post-COVID, *n* (%)	12 (7.6)	7 (8.0)	0.01	0.92
Sicca Syndrome Post-COVID, *n* (%)	19 (12.0)	9 (10.2)	0.18	0.67
Raynaud’s phenomenon Post-COVID, *n* (%)	1 (0.6)	0 (0)	0.56	0.45
Myalgia Post-COVID, *n* (%)	63 (39.9)	30 (34.1)	0.80	0.37
**Exertional dyspnea Post-COVID, *n* (%)**	127 (80.4)	57 (64.8)	7.30	**0.01**
Chest pain Post-COVID, *n* (%)	28 (17.7)	16 (18.2)	0.01	0.93
Sore throat Post-COVID, *n* (%)	4 (2.5)	5 (5.7)	1.59	0.21
Sputum Post-COVID, *n* (%)	12 (7.6)	7 (8.0)	0.01	0.92
Rhinitis Post-COVID, *n* (%)	8 (5.1)	8 (9.1)	1.51	0.22
Loss of appetite Post-COVID, *n* (%)	10 (6.3)	5 (5.7)	0.04	0.84
Days from COVID onset, mean ± SD	173.38 ± 96.78	155.42 ± 88.23	2.07	0.15
SpO_2_, mean± SD	96.83 ± 5.46	96.67 ± 3.69	5902	0.41
HAM-D, mean ± SD	3,74 ± 4.25	3.89 ± 4.48	6764	0.72
HAM-A, mean ± SD	4.78 ± 4.96	4.86 ± 6.52	6409	0.31
YMRS, mean ± SD	0.89 ± 1.44	0.89 ± 1,43	6810	0.77
KMRSD, mean ± SD	6.49 ± 2.32	6.65 ± 2.38	6949	0.99
BPRS, mean ± SD	26.51 ± 4.70	26.59 ± 3.46	6631	0.53
CAPS Total; mean ± SD	6.18 ± 9.72	7.74 ± 10.89	6525	0.41
CD-RISC Total; mean ± SD	67.88 ± 13.38	70.38 ± 12.57	6032	0.09
**DERS Total; mean ± SD**	83.08 ± 15.06	78.32 ± 15.56	5471	**<0.01**
SHAPS Total, mean ± SD	0.82 ± 2.25	.53 ± 2.02	6339	0.12
PSQI Total; mean ± SD	6.24 ± 5.33	5.76 ± 4.94	6659	0.58
BHS Total; mean ± SD	7.99 ± 4.90	7.41 ± 4.89	6193	0.15

Legend: significant results are in bold. Abbreviations: Anti-IL-6-R: patients who received anti-interleukin-6-receptor treatment; BHS, Beck Hopelessness Scale; BPRS, Brief -Psychiatric Rating Scale; CAPS, Clinician-Administered PTSD Scale; CD-RISC, Connor–-Davidson Resilience Scale; Ctrl: patients who did not receive anti-interleukin-6-receptor; DERS, Difficulties in Emotion Regulation scale; HAM-A, Hamilton Anxiety rating scale; HAM-D, Hamilton rating scale for depression; KMDRS, Koukopoulos Miixed Depression Rating Scale; PSQI, Pittsburgh Sleep Quality Index; SD, standard deviation; SHPS, Snaith–-Hamilton Pleasure Scale; SpO_2_, peripheral capillary oxygen saturation.

**Table 3 brainsci-14-00047-t003:** Differences in sociodemographic, clinical history prior to COVID-19, and COVID-19 characteristics among TOC (*N* = 67), SAR (*N* = 21), and Ctrl (*N* = 158).

		Ctrl	TOC	SAR	χ^2^	*p*	Post hoc		
							Ctrl vs. TOC	Ctrl vs. SAR	TOC vs. SAR
**Sociodemographic and clinical characteristics**									
**Age (y), mean ± SD**	**56.66 ± 10.30**		**56.00 ± 13.03**	**59.76 ± 8.21**	1.34	0.51	0.91	0.25	0.32
BMI, mean ± SD	28.00 ± 4.57		28.19 ± 4.91	28.35 ± 3.83	0.48	0.79	0.86	0.49	0.59
**Female, *n* (%)**	58 (36.7)		24 (35.8)	2 (9.5)	6.21	**0.04**	0.90	**0.01**	**0.02**
Marital Status (%)									
Never married	18 (11.5)		12 (17.9)	3 (14.3)	6.08	0.64	0.19	0.7	0.7
Married	101 (63.9)		45 (67.2)	16 (76.2)			0.64	0.27	0.44
Partner	7 (4.4)		2 (3)	0 (0)			0.61	0.33	0.42
Divorced	7 (4.4)		2 (3)	0 (0)			0.61	0.33	0.42
Widowed	25 (15.8)		6 (9)	2 (9.5)			0.17	0.45	0.94
Occupation (%)									
Unemployed		9 (5.7)	6 (9)	1 (4.8)	3.01	0.81	0.37	0.86	0.54
Employed		117 (74.1)	48 (71.6)	14 (66.7)			0.71	0.47	0.66
Retired		32 (20.3)	13 (19.4)	6 (28.8)			0.88	0.38	0.37
Education (y), mean ± SD		12.80 ± 3.93	13.58 ± 4.14	14.35 ± 4.16	1.86	0.16	0.21	0.07	.31
Comorbidities, mean ± SD		2.69 ± 2.40	2.43 ± 2.46	2.14 ± 1.74	1.59	0.45	0.28	0.39	0.96
Psychiatric treatments prior to COVID, mean ± SD		0.07 ± 0.30	0.09 ± 0.38	0.14 ± 0.36	2.15	0.34	0.91	0.15	0.25
Pneumological comorbidities, *n* (%)		31(19.6)	7(10.4)	1 (4.8)	5.08	0.08	0.09	0.09	0.43
**COVID-19**									
Antibiotics prior to admission, *n* (%)		89 (56.3)	43 (64.2)	9 (42.9)	3.15	0.21	0.27	0.24	0.08
**Cortisone prior to admission, *n* (%)**		67 (42.4)	23 (34.3)	3 (14.3)	6.71	**0.03**	0.26	**0.01**	0.08
Drug treatments prior to admission (*n*), mean ± SD		1.27 ± 1.64	0.99 ± 1.49	1.57 ± 2.13	2.58	0.28	0.31	0.17	0.41
Hospitalization (d), mean ± SD		22.35 ± 25.1	24.75 ± 19.38	31.95 ± 20.74	1.64	0.20	0.76	0.18	0.43
Psychiatric symptoms/delirium during hospitalization, *n* (%)		8 (5.1)	8 (11.9)	3 (14.3)	4.49	0.11	0.07	0.10	0.78
Psychiatric treatments during hospitalization (*n*), mean ± SD		0.16 ±0.46	0.21 ± 0.54	0.10 ± 0.30	0.55	0.76	0.62	0.65	0.49
**Oxygen therapy, *n* (%)**		141 (89.2)	55 (82.1)	14 (66.7)	8.35	**0.01**	0.14	**<0.01**	0.13
Drugs during hospitalization(*n*), mean± SD		2.46 ± 2.56	2.18 ± 2.5	1.67 ± 2.03	1.05	0.35	0.45	0.18	0.42
Intensive Care Unit admission, *n* (%)		113 (71.5)	46 (68.7)	11 (52.4)	3.19	0.2	0.67	0.07	0.17
Symptoms during COVID, mean ± SD		7.59 ± 3.96	6.81 ± 3.63	6.62 ± 2.82	2.63	0.29	0.15	0.31	0.83
Fever, *n* (%)		142 (89.9)	60 (89.6)	21 (100.0)	2.37	0.30	0.94	0.13	0.12
**Fatigue/Asthenia, *n* (%)**		133 (84.7)	46 (68.7)	16 (76.2)	7.61	**0.02**	**<0.01**	0.36	0.51
Cough, *n* (%)		97 (61.4)	41 (61.2)	15 (71.4)	0.83	0.66	0.97	0.37	0.39
Diarrhea, *n* (%)		40 (25.3)	13 (19.4)	3 (14.3)	1.87	0.39	0.34	0.27	0.60
**Headache, *n* (%)**		81 (51.3)	26 (38.8)	4 (19.0)	9.25	**0.01**	0.09	**<0.01**	0.97
Anosmia/Dysosmia, *n* (%)		62 (39.2)	32 (47.8)	7 (33.3)	1.98	0.37	0.23	0.60	0.25
Dysgeusia, *n* (%)		65 (41.1)	31 (46.3)	9 (42.9)	0.51	0.78	0.48	0.88	0.79
Red eyes, *n* (%),		27 (17.1)	10 (14.9)	4 (19.0)	0.25	0.88	0.69	0.83	0.65
Reduction in eyesight, *n* (%)		25 (15.8)	9 (13.4)	3 (14.3)	0.22	0.90	0.65	0.86	0.92
Syncope, *n* (%),		15 (9.5)	5 (7.5)	2 (9.5)	0.25	0.88	0.62	1	0.76
Vertigo, *n* (%)		46 (29.1)	12 (17.9)	3 (14.3)	4.53	0.10	0.08	0.15	0.69
Arthralgia, *n* (%)		95 (60.1)	33 (49.3)	9 (42.9)	3.79	0.15	0.13	0.13	0.61
**Cutaneous and mucosal lesions, *n* (%)**		25 (15.8)	4 (6.0)	0 (0.0)	7.46	**0.02**	**0.04**	**0.04**	0.25
Sicca Syndrome, *n* (%)		38 (24.1)	11 (16.4)	3 (14.3)	2.29	0.32	0.20	0.32	0.82
Loss of appetite, *n* (%)		69 (43.9)	35 (52.2)	12 (57.1)	2.18	0.34	0.24	0.24	0.70
Raynaud’s phenomenon, *n* (%)		2 (1.3)	0 (0.0)	0 (0.0)	1.12	0.57	0.35	0.60	0.9999
Myalgia, *n* (%)		95 (60.1)	29 (43.3)	10 (47.6)	5.82	0.06	**0.02**	0.28	0.73
Chest Pain, *n* (%)		56 (35.4)	19 (28.4)	6 (28.6)	1.27	0.53	0.30	0.54	0.98
Sore throat, *n* (%)		35 (22.2)	16 (23.9)	5 (23.8)	0.09	0.95	0.78	0.87	0.99
Sputum, *n* (%)		32 (20.3)	13 (19.4)	4 (19.0)	0.03	0.98	0.88	0.90	0.97
Rhinitis, *n* (%)		20 (12.7)	11 (16.4)	3 (14.3)	0.56	0.75	0.45	0.83	0.82

Legend: significant results are in bold. Abbreviations: Ctrl, patients who did not receive anti-interleukin-6-receptor SAR, patients receiving sarilumab during hospitalization; SD, standard deviation; TOC, patients receiving tTocilizumab during hospitalization.

**Table 4 brainsci-14-00047-t004:** Differences in clinical features and scores on psychopathological scales at post-COVID-19 evaluation between TOC (*N* = 67), SAR (*N* = 21), and Ctrl (*N* = 158).

	Ctrl (*N* = 158)	TOC (*N* = 67)	SAR (*N* = 21)	χ^2^	*p*	*Post hoc*		
						Ctrl vs. TOC	Ctrl vs. SAR	TOC vs. SAR
**Post-COVID symptoms, mean ± SD**	3.88 ± 2.78	3.24 ± 2.59	3.19 ± 2.80	6.27	**0.04**	0.08	**0.04**	0.38
Post-COVID Fatigue/Asthenia, *n* (%)	102 (64.6)	37 (55.2)	13 (61.9)	1.74	0.42	0.19	0.81	0.59
Post-COVID Cough, *n* (%)	28 (17.7)	4 (6.0)	2 (9.5)	5.81	0.06	**0.02**	0.35	0.58
Post-COVID Diarrhea, *n* (%)	7 (4.4)	0 (0.0)	0 (0.0)	4.01	0.13	0.08	0.33	0.999
Post-COVID Headache, *n* (%)	35 (22.2)	7 (10.4)	2 (9.5)	5.48	0.06	**0.04**	0.18	0.90
Post-COVID Anosmia/dysosmia, *n* (%)	17 (10.8)	8 (11.9)	1 (4.8)	0.89	0.64	0.79	0.39	0.34
Post-COVID Dysgeusia, *n* (%)	22 (13.9)	6 (9.0)	1 (4.8)	2.20	0.33	0.30	0.24	0.53
Post-COVID Red Eyes, *n* (%)	8 (5.1)	5 (7.5)	0 (0.0)	1.82	0.40	0.38	0.20	0.20
Post-COVID Reduction in eyesight, *n* (%)	26 (16.5)	10 (14.9)	6 (28.6)	2.22	0.33	0.77	0.17	0.16
Post-COVID Syncope, *n* (%)	0 (0.0)	0 (0.0)	0 (0.0)	–	–	–	–	–
Post-COVID Vertigo, *n* (%)	22 (13.9)	5 (7.5)	2 (9.5)	2.0	0.37	0.17	0.58	0.76
Post-COVID Arthralgia, *n* (%)	62 (39.2)	23 (34.3)	8 (38.1)	0.48	0.78	0.49	0.92	0.75
Post-COVID Cutaneous and mucosal lesions, *n* (%)	12 (7.6)	6 (9.0)	1 (4.8)	0.40	0.82	0.73	0.64	0.54
Post-COVID Sicca Syndrome, *n* (%)	19 (12.0)	7 (10.4)	2 (9.5)	0.19	0.91	0.73	0.74	0.90
Post-COVID Raynaud’s phenomenon, *n* (%)	1 (0.6)	0 (0.0)	0 (0.0)	0.56	0.76	0.52	0.71	0.999
Post-COVID Myalgia, *n* (%)	63 (39.9)	22 (32.8)	8 (38.1)	0.99	0.61	0.32	0.87	0.66
**Post-COVID Dyspnea on exertion, *n* (%)**	127 (80.4)	46 (68.7)	11 (52.4)	9.55	**0.01**	0.06	**<0.01**	0.17
Post-COVID Chest Pain, *n* (%)	28 (17.7)	12 (17.9)	4 (19.0)	0.02	0.99	0.97	0.88	0.90
Post-COVID Sore Throat, *n* (%)	4 (2.5)	5 (7.5)	0 (0.0)	4.12	0.13	0.08	0.46	0.20
Post-COVID Sputum, *n* (%)	12 (7.6)	5 (7.5)	2 (9.5)	0.11	0.95	0.98	0.76	0.76
Post-COVID Rhinitis, *n* (%)	8 (5.1)	6 (9.0)	2 (9.5)	1.52	0.47	0.27	0.40	0.94
Post-COVID Loss of Appetite, *n* (%)	10 (6.3)	3 (4.5)	2 (9.5)	0.75	0.69	0.59	0.58	0.38
Days from COVID Symptoms Onset, *n* (%)	173.38 ± 96.78	152.96 ± 90.13	163.29 ± 83.48	1.13	0.32	0.30	0.89	0.90
SpO_2_, mean ± SD	96.83 ± 5.46	96.68 ± 4.22	96.62 ± 1.12	0.74	0.69	0.41	0.49	0.86
BPRS Total, mean ± SD	26.51 ± 4.70	26.49 ± 3.62	26.90 ± 2.95	1.74	0.42	0.29	0.55	0.25
HAM-A Total, mean ± SD	4.78 ± 4.96	4.82 ± 6.13	5.00 ± 7.82	1.15	0.56	0.42	0.39	0.76
HAM-D Total, mean ± SD	3.74 ± 4.25	4.03 ± 4.4	3.43 ± 4.83	0.57	0.75	0.56	0.75	0.47
MRS Total, mean ± SD	0.89 ± 1.44	0.85 ± 1.39	1.00 ± 1.58	0.09	0.96	0.80	0.84	0.94
KMRSD Total, mean ± SD	6.49 ± 2.32	6.72 ± 2.19	6.43 ± 2.96	1.64	0.44	0.59	0.28	0.23
CAPS Total, mean ± SD	6.18 ± 9.72	7.46 ± 10.57	8.62 ± 12.09	1.04	0.60	0.61	0.34	0.54
CD-RISC Total, mean ± SD	67.88 ± 13.38	69.94 ± 12.64	71.76 ± 12.54	3.04	0.22	0.15	0.65	0.23
**DERS Total, mean ± SD**	83.08 ± 15.06	77.06 ± 14.92	82.33 ± 17.21	9.58	**<0.01**	**<0.01**	0.62	0.14
SHAPS Total, mean ± SD	0.82 ± 2.25	0.58 ± 2.26	0.38 ± 0.97	2.39	0.30	0.15	0.44	0.89
PSQI Total, mean ± SD	6.24 ± 5.33	5.79 ± 5.21	5.67 ± 4.07	0.34	0.84	0.57	0.86	0.81
BHS Total, mean ± SD	8.16 ± 4.78	7.00 ± 5.06	8.71 ± 4.14	4.38	0.11	0.05	0.68	0.12

Legend: significant results are in bold. *Abbreviations:* BHS, Beck Hopelessness Scale; BPRS, Brief Psychiatric Rating Scale; CAPS, Clinician-Administered PTSD Scale, CD-RISC, Connor–Davidson Resilience Scale; Ctrl, patients who did not receive anti-interleukin-6-receptor treatment; DERS, Difficulties in Emotion Regulation scale; HAM-A, Hamilton Anxiety rating scale; HAM-D, Hamilton rating scale for depression; KMDRS, Koukopoulos Mixed Depression Rating Scale; PSQI, Pittsburgh Sleep Quality Index; SAR, patients receiving sarilumab during hospitalization; SD, standard deviation; SHAPS, Snaith–Hamilton Pleasure Scale; SpO_2_, peripheral capillary oxygen saturation; TOC, patients receiving Tocilizumab during hospitalization.

## Data Availability

The deidentified data presented in this study are available on request from the corresponding author upon reasonable request. The data are not publicly available due to privacy issues.

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
