# Peer review of "Effect of Anti-Interleukin-6 Agents on Psychopathology in a Sample of Patients with Post-COVID-19 Syndrome: An Observational Study"

_brainsci, 2024, doi:10.3390/brainsci14010047_

Round 1

Reviewer 1 Report

Comments and Suggestions for Authors

Dear Authors

This article reported the effect of anti-interleukin-6 agents on psychopathology in a sample of patients with post-COVID-19 syndrome, which is an interesting topic today and well done. However, some concerns in this article need to be addressed.

1.      Please remove the p-value in the abstract section.

2.      In the keywords section, please change "post-COVID-19 syndrome" to " Post-COVID-19 syndrome ".

3.      In the first citation, put the full name before the word abbreviation, followed by the abbreviation (e.g. COVID-19, FDA). Please correct them.

4.      The introduction section is too long. Please summarize it.

5.      In line 66, please change long-COVID to Long COVID.

6.      Please enter your questionnaire as a supplementary file.

7.      Did these people have a history of COVID-19 vaccination?

8.      I propose to present some data in the form of graph.

9.      Please remove a 3.2 section.

10.   How long after the COVID-19 infection were the patients included in this study?

11.   Please discuss further and improve the discussion section.

12.   Research suggestion was not addressed at the end of the discussion. Please include it.

13.   Please include the acknowledge section.

Comments on the Quality of English Language

Minor editing of English language required

Author Response

Dear Authors

This article reported the effect of anti-interleukin-6 agents on psychopathology in a sample of patients with post-COVID-19 syndrome, which is an interesting topic today and well done. However, some concerns in this article need to be addressed.

We thank Reviewer 1 for having appreciated the importance of our study. Please find or changes in the manuscript in red characters.

  1. Please remove the p-value in the abstract section.

We removed p values from the Abstract.

  1. In the keywords section, please change "post-COVID-19 syndrome" to " Post-COVID-19 syndrome ".

We changed "post-COVID-19 syndrome" to " Post-COVID-19 syndrome " in the keyword section

  1. In the first citation, put the full name before the word abbreviation, followed by the abbreviation (e.g. COVID-19, FDA). Please correct them.

We apologize for the oversight and corrected all the citations.

  1. The introduction section is too long. Please summarize it.

We understand Reviewer 1 comment. We tried to shrink the section, while still maintaining the meaning. We shortened it as much as possible, even though the bulk of the introduction remained unchanged, as all concepts developed in this section were instrumental in understanding the work.

  1. In line 66, please change long-COVID to Long COVID.

We changed the word long-COVID to Long COVID in line 66.

  1. Please enter your questionnaire as a supplementary file.

We added questionnaires used in the Supplement.

  1. Did these people have a history of COVID-19 vaccination?

We thank Reviewer 1. The sample enrolled contracted the COVID-19 prior to the availability of vaccines. We added a statement on this aspect in the result section.

  1. I propose to present some data in the form of graph.

We thank Reviewer 1 for this suggestion, we made a boxplot regarding between group differences at DERS.

  1. Please remove a 3.2 section.

We removed section 3.2

  1. How long after the COVID-19 infection were the patients included in this study?

We Thank Reviewer 1 for this question. Timeframe between COVID-19 and enrollment vary. We gave this information in Table 2 and Table 4.

  1. Please discuss further and improve the discussion section.

We thank Reviewer 2 for the suggestion. We expanded the discussion section. Specifically, we explained more in detail possible causes underlying between-group differences in DERS, which represents the most important finding regarding psychopathological scales.

  1. Research suggestion was not addressed at the end of the discussion. Please include it.

We added a statement on research suggestion and future directions, however, we decided to add it in the Conclusion section.

  1. Please include the acknowledge section.

We added an acknowledgment section at the end of the manuscript.

Reviewer 2 Report

Comments and Suggestions for Authors

The manuscript entitled “Effect of anti-interleukin-6 agents on psychopathology in a sample of patients with post-COVID-19 syndrome. An observational study” reports a possible effect of the use of IL-6 receptor inhibitors Tocilizumab and Sarilumab on improving psychological and neurological symptoms during post-COVID-19 syndrome. The manuscript is a well-written and well-structured article, which was based on the analysis of multiple socio-demographic and clinical data, as well as neurological symptoms, and different psychological inventories, which were used to measure depression, anxiety, PTSD, resilience, sleep problems, etc. However, several issues need to be addressed.

1.     I suggest adding the information on sex ratio, as well as age range in the Abstract.

2.     Please report in details the symptoms, which are characteristic for post-COVID-19 syndrome in the Introduction and duration of their existence if it was analyzed. In addition, it is significant to report a period after receiving a negative COVID-19 test, in which psychological measurements were performed.

3.     Please indicate in the Introduсtion that Actemra and Tocilizumab represent the same drug.

4.     The authors have to give more details about the “as-usual schedule of treatment” characteristic for the control group. Moreover, please, provide more detailed explanation on the “Comorbidities prior to COVID-19”.

5.     I have some comments on the Statistical analysis. First, the authors used Student’s t-test/ANOVA; however, they have not provided the data on the correspondence of their quantitative data (values from psychological inventories) to the Gaussian distribution. Since the sample size is rather small, I can suppose that the distribution will be abnormal in some or all quantitative scales. Please, provide corresponding statistical criteria, which can help to make a conclusion on the appropriateness of the selected parametric tests. In addition, it would be of interest to provide a correlation table, since there are multiple psychological measures.

6.     The authors stated (on line 337) “Ctrl had more patients who required oxygen-therapy during acute COVID-19 infection.” It seems that it can distort the effect of using anti-IL-6-R drugs on psychological characteristics.

7.     Please, report what is given in brackets in Table 1. If it is a proportion, it seems that proportion for “Never married” is given incorrectly. Also, please indicate that the bold letters stand for statistically significant differences in Table 1. It seems that there was a borderline difference between Ctrl and Anti-IL-6_R group in Cortisone treatment prior to admission.  Please, make similar explanations in notes to other Tables.

8.     I suggest changing the terms in the titles in Tables (i.e. psychopathological variables) and such mentioning in the text, where the authors report the values for neurological variables and psychological characteristics. Psychopathology includes other DSM-V clinical symptoms, that are examined by psychiatrists to diagnose psychopathology. I.e. on line 193 there is a mention of “… presence of psychopathology during the post-COVID-19 syndrome as assessed with rating scales.”. It’s incorrect, psychopathology cannot be assessed via inventories only. Please, check the text for such discrepancy.

Comments on the Quality of English Language

Minor editing of English is required.

Author Response

The manuscript entitled “Effect of anti-interleukin-6 agents on psychopathology in a sample of patients with post-COVID-19 syndrome. An observational study” reports a possible effect of the use of IL-6 receptor inhibitors Tocilizumab and Sarilumab on improving psychological and neurological symptoms during post-COVID-19 syndrome. The manuscript is a well-written and well-structured article, which was based on the analysis of multiple socio-demographic and clinical data, as well as neurological symptoms, and different psychological inventories, which were used to measure depression, anxiety, PTSD, resilience, sleep problems, etc. However, several issues need to be addressed.

We thank Reviewer 2 for having appreciated the importance of our study. Please find or changes in the manuscript in red characters

  1. I suggest adding the information on sex ratio, as well as age range in the Abstract.

We added sex ratio and age range in the Abstract

  1. Please report in details the symptoms, which are characteristic for post-COVID-19 syndrome in the Introduction and duration of their existence if it was analyzed. In addition, it is significant to report a period after receiving a negative COVID-19 test, in which psychological measurements were performed.

                  We reported in details the post-COVID-19 syndrome symptoms’ duration in the Introduction. We do not information regarding negativization, but only the timeframe regarding symptoms onset and psychopathological measurements. We mentioned this lack in the Limitations section.  

  1. Please indicate in the Introduсtion that Actemra and Tocilizumab represent the same drug.

We specified that Actemra and Tocilizumab represent the same drug in the introduction section.

  1. The authors have to give more details about the “as-usual schedule of treatment” characteristic for the control group. Moreover, please, provide more detailed explanation on the “Comorbidities prior to COVID-19”.

                  We thank Reviewer 2 for highlighting such lack. Usual schedule is composed by corticosteroids and heparin, plus drugs rlated to each subject’s comorbidity. We highlighted this aspect at the end of the Introduction. Also, a better explanation on comorbidities was given in the “Assessment” section.

  1. I have some comments on the Statistical analysis. First, the authors used Student’s t-test/ANOVA; however, they have not provided the data on the correspondence of their quantitative data (values from psychological inventories) to the Gaussian distribution. Since the sample size is rather small, I can suppose that the distribution will be abnormal in some or all quantitative scales. Please, provide corresponding statistical criteria, which can help to make a conclusion on the appropriateness of the selected parametric tests. In addition, it would be of interest to provide a correlation table, since there are multiple psychological measures.

We thank Reviewer 2 for such comment. Although all the data distribution was skewed, we preferred using parametric tests instead a non-parametric tests. Reasons behind this choice is to exploit the parametrics’ test statistical power, which is higher than the one of non-parametric tests. Such choice was also encouraged by the groups’ sample size, though not large, but over the minimum threshold required to perform these analyses. However, to strengthen the analyses’ reliability, we agreed with the Reviewer 2 point of view and performed non-parametric analyses for the whole manuscript. We amended the manuscript accordingly. Changes made are present in methods and results sections, and in Tables. We also provided a correlation table in which relationships among scale were explored. In order to ease the manuscript readability, this table has been placed as a supplementary material. 

  1. The authors stated (on line 337) “Ctrl had more patients who required oxygen-therapy during acute COVID-19 infection.” It seems that it can distort the effect of using anti-IL-6-R drugs on psychological characteristics.

We thank Reviewer 2 for this comment. In order to reduce the possible effect of this variable, as well as others differentiating controls from subjects under anti-IL-6-R medications, we corrected our analyses for the aforementioned variables. As a consequence, results are not affected by the varibles included in the model. However, we agree that oxygen-therapy might be a determinant confounding factor. We acknowledged this aspect in Limitations.

  1. Please, report what is given in brackets in Table 1. If it is a proportion, it seems that proportion for “Never married” is given incorrectly. Also, please indicate that the bold letters stand for statistically significant differences in Table 1. It seems that there was a borderline difference between Ctrl and Anti-IL-6_R group in Cortisone treatment prior to admission.  Please, make similar explanations in notes to other Tables.

We thank Reviewer 2 for this oversight. We corrected mistakes and oversights in the tables and made explanation notes more homogenous. Sometimes, some authors highlighted trend-level-differences in the tables. We decided to not choose this option in order to give a clear-cut between what is significant and what is not not, given all the amounts of variables analyzed in the text.

  1. I suggest changing the terms in the titles in Tables (i.e. psychopathological variables) and such mentioning in the text, where the authors report the values for neurological variables and psychological characteristics. Psychopathology includes other DSM-V clinical symptoms, that are examined by psychiatrists to diagnose psychopathology. I.e. on line 193 there is a mention of “… presence of psychopathology during the post-COVID-19 syndrome as assessed with rating scales.”. It’s incorrect, psychopathology cannot be assessed via inventories only. Please, check the text for such discrepancy.

We thank Reviewer 2 for this comment. We changed the term “psychopathology” with “psychiatric symptoms”. On the other hand, sometimes we maintained the term “psychopathology”. For instance, scales used regard psychopathology, as well as the BPRS is used to assess general psychopathology. Since subjects underwent an assessment composed by both rating scales and an extensive interview made by psychiatrists, we maintained the term “psychopathology” in the title.

Reviewer 3 Report

Comments and Suggestions for Authors

The research conducted by the authors of the manuscript is undoubtedly significant and relevant, especially during the period of active Covid-19 infection. However, it would make sense for the authors to show the study design more clearly, perhaps using flowcharts. It also remains unclear whether there was any comorbidity and whether the patients received additional treatment in combination with the anti-Covid-19 drugs described by the authors. It is also not entirely clear whether the mental status of patients before illness was assessed (possibly based on interviews with relatives), this would help to more clearly determine the duration of the onset of these disorders and their relationship with Covid-19. In the “conclusions” section, the authors need to more clearly describe the possible application of their results in clinical practice, which could significantly increase the significance of the manuscript and the interest of readers.

Author Response

The research conducted by the authors of the manuscript is undoubtedly significant and relevant, especially during the period of active Covid-19 infection. However, it would make sense for the authors to show the study design more clearly, perhaps using flowcharts. It also remains unclear whether there was any comorbidity and whether the patients received additional treatment in combination with the anti-Covid-19 drugs described by the authors. It is also not entirely clear whether the mental status of patients before illness was assessed (possibly based on interviews with relatives), this would help to more clearly determine the duration of the onset of these disorders and their relationship with Covid-19. In the “conclusions” section, the authors need to more clearly describe the possible application of their results in clinical practice, which could significantly increase the significance of the manuscript and the interest of readers.

We thank Reviewer 3 for his/her comments. We agree with Reviewer 3 comment on the lack of clarity regarding the study design. We better explained this aspect in Material and Methods. As regard medical comorbidities, no differences emerged, as described in Table 1 and Table 3. As regarded psychiatric comorbidity, only four patients had a comorbidity (2 with comorbid anxiety, 1 with comorbid personality disorder, 1 with obsessive compulsive disorder, two belonging to the Tocilizumab arm, and one belonging to the controls. Since the subjects’ number is too low, we preferred to not perform analyses of these comorbidities. As regards subjects with anti IL-6-R medications, additional medications include enoxaparin and dexamethasone, plus other medications related to subjects’ comorbidities. We added a statement on this aspect at the end of the Introduction. The total number of drugs assumed is already present in Table 1. The subjects’ mental status was assessed through a detail psychiatric anamnesis performed by a team’s psychiatrist and then confirmed by the subjects’ treating psychiatrist (whenever present) and/or an interview with relatives. We gave an explanation of this process in Material and Methods. As regards Conclusions, we added a section describing possible clinical applications of the results found.  

Round 2

Reviewer 2 Report

Comments and Suggestions for Authors

The authors have addressed all my previous issues. The article is suitable for publication.